# Impact of Temperature Change on the Fall Armyworm, *Spodoptera frugiperda* under Global Climate Change

**DOI:** 10.3390/insects13110981

**Published:** 2022-10-26

**Authors:** Xiao-Rui Yan, Zhen-Ying Wang, Shi-Qian Feng, Zi-Hua Zhao, Zhi-Hong Li

**Affiliations:** 1MARA Key Laboratory of Surveillance and Management for Plant Quarantine Pests, College of Plant Protection, China Agricultural University, Beijing 100193, China; yanxr1120@126.com (X.-R.Y.); zhzhao@cau.edu.cn (Z.-H.Z.); 2Sanya Institute of China Agricultural University, Sanya 572025, China; 3State Key Laboratory for Biology of Plant Diseases and Insect Pests, Institute of Plant Protection, Chinese Academy of Agricultural Sciences, Beijing 100193, China; wangzhenying@caas.cn (Z.-Y.W.); fengshiqian@163.com (S.-Q.F.)

**Keywords:** *Spodoptera frugiperda*, fluctuating temperature, temperature tolerance, pest management program

## Abstract

**Simple Summary:**

The fall armyworm (FAW) is of tropical–subtropical origin and defined as one of the most destructive agricultural pests globally. Superior migratory performance, reproductive ability and adaptability make it successful in causing a serious loss to agricultural production. Since this species lacks a diapause mechanism, temperature influences the population dynamic of the FAW to a great extent and changes metabolic and developmental states as a result, indirectly affecting the degree of crop infested. Control technologies can be put forward comprehensively in consideration of the effects of temperature on the FAW. In this review, we discussed the biological manifestation and tolerance of the FAW with various temperatures and proposed constructive suggestions for controlling this species and future research direction. This information is valuable for understanding the relationships between insect pests and temperature, strengthening the monitoring and pest control, providing service and support for newly developed strategies in the near future.

**Abstract:**

The fall armyworm (FAW), *Spodoptera frugiperda* (J. E. Smith, 1797), known as an important agricultural pest around the world, is indigenous to the tropical–subtropical regions in the Western Hemisphere, although its distribution has expanded over large parts of America, Africa, Asia and Oceania in the last few years. The pest causes considerable costs annually coupled with its strong invasion propensity. Temperature is identified as the dominant abiotic factor affecting herbivorous insects. Several efforts have reported that temperature directly or indirectly influences the geographic distribution, phenology and natural enemies of the poikilothermal FAW, and thus may affect the damage to crops, e.g., the increased developmental rate accelerates the intake of crops at higher temperatures. Under some extreme temperatures, the FAW is likely to regulate various genes expression in response to environmental changes, which causes a wider viability and possibility of invasion threat. Therefore, this paper seeks to review and critically consider the variations of developmental indicators, the relationships between the FAW and its natural enemies and the temperature tolerance throughout its developmental stage at varying levels of heat/cold stress. Based on this, we discuss more environmentally friendly and economical control measures, we put forward future challenges facing climate change, we further offer statistical basics and instrumental guidance significance for informing FAW pest forecasting, risk analyses and a comprehensive management program for effective control globally.

## 1. Introduction

Climate change has been occurring in the 21st century and the result of such changes may raise the loss of host plants and increase the risks to food security because of the occurrence and spread of insect pest herbivores, great challenge will arise for biologists due to the consequence on natural ecosystems. Climate change can impact directly or indirectly, through other factors, notably the host plant and natural enemy resources, the physiology and behavior of insect pests. At the temperature optimum, survivability, fecundity and development can perform well [1,2]. Consequently, a temperature value either higher or lower than the optimum temperature has more noteworthy adverse effects on insect populations [3]. Although insects can survive under low temperature, resulting in immature gonads in adults and less fecundity, sustained heat breaks the mitochondria of insect cells, enzymes, hormone activities and the mating behavior of adults [4]. Much progress has been made in documenting shifts in insect pest development parameters and distributions in response to temperature change scenarios and scholars have proposed several efficient models of the developmental rate–temperature relationship of insect pests [5,6,7]. Understanding how insect pests respond to changing temperature is crucial in the context of climate change and can steer the decision-making to the rationalization of the anticipatory management principle in order to minimize pest infestation levels [8].

The fall armyworm (FAW), *Spodoptera frugiperda* (J. E. Smith, 1797) (Lepidoptera, Noctuidae), is an increasingly essential laboratory and field model organism in biological and agricultural research considering its widespread and expanding dispersal and financial and socioeconomic relevance. The FAW is an extremely destructive omnivorous pest of subtropical and tropical origin with higher viabilities over a wide range of temperatures and distributions [9]. Its strong fertility ability, high migratory capacity and ecological plasticity contribute to the FAW’s major economic damage by voraciously infiltrating key growing areas of at least 353 known different host plant species belonging to 76 botanical families, e.g., corn (*Zea mays* L.), rice, sorghum, sugarcane, cotton and varieties of vegetables [10]. Without any effective control methods, the yield losses are expected to reach 8.3 million to 20.6 million tons in 12 main African corn-producing countries, 32% in Ethiopia and 47% in Kenya based on socioeconomic surveys [11,12]. Corn is the first grain crop and has been widely grown and maintained in China [13], the infested area of corn by the FAW has reached 1.14 billion m^2^ by May 2022 (National Agro-Tech Extension and Service Center, NATESC), ranging from the eastern boundary (33.4° N) to the western point (31.6° N) [14,15,16]; without strict supervision measures for the FAW, it would pose a serious threat to the safe production of food crops in China.

Analyses from the literature have revealed that growth (passing through all life stages) and development rates of the FAW have a temperature dependence. This ectothermic noctuid species does not undergo diapause, so it cannot survive extended periods of inhospitable conditions including extreme cold temperature [17]. The effect of temperature change on the FAW can be direct, through impacts on their distribution, physiology and flight performance, or indirect, through other factors, notably host plants and natural enemies. At the same time, existing studies have showed that the FAW can maintain cellular homeostasis by regulating gene expression in response to extreme environmental stress. In this review, we summarize previously published and current data concerning the relevant effects of changing temperature (as a principal representative of climate change) on the FAW population distribution, phenology, natural enemies and considering the mechanisms of temperature tolerance, hoping to come up with effective control strategies and make reasonable predictions as to their seasonal and phenological occurrence and migration and provide support of urgent monitoring and management for preventing the areawide invasion of *S. frugiperda*, which will enhance the safety of global agriculture.

## 2. Emergence and Distribution of Fall Armyworm

The FAW was firstly discovered as an injurious pest in 1797 in Georgia [18]. From 1856 to 1928, numerous events of “marching-worms” outbreaks were recorded in the U.S. [19]. After 1928, the spreads of the FAW were not well documented until the FAW became a predominant pest and devastated crops in some southern areas. Subsequently, it was confirmed that it had proliferated throughout almost all 44 countries of sub-Sahara including Nigeria, Benin, Togo, Ghana, São Tomé and Princípe in Africa and in Asian countries such as India, Thailand, Bangladesh and Myanmar in 2016–2018 [20,21,22,23]. On 11 January 2019, the initial detection of *S. frugiperda* larval infestation was in a cornfield of Jiangcheng county, Yunnan province, China, then, except for Xinjiang, Qinghai and Northeastern China, it was confirmed in 26 provinces (autonomous regions, municipalities) by September 2019 [24,25,26,27,28].

In China, a large area (south of 28° N, corresponding with the 10 °C isotherm in January) is highly suitable for year-round occurrence of the FAW and a partial area (28° N to 31° N, corresponding with the 6 °C and 10 °C isotherm in January) is conditional for the winter-breeding of the FAW [29,30,31,32]. Lacking a diapause trait, this species must begin a new series of migratory flights when conditions are not suitable for survival [17,33]. With an elevated temperature, not only is the horizontal occurrence range expected to extend widely, but also latitudes and altitudes will move higher among the range of activities of the majority of populations; contrary to what was previously thought, the number of insect species per mean area decreased with increasing latitude and altitude at normal temperatures.

A species distribution model (SDM) is a kind of niche measuring model, which studies the environmental tolerance of species based on the known distribution points of organisms and their related environmental factors. An SDM can predict a species distribution range under future climate scenarios, which is beneficial to facilitate forewarning of this highly notorious pest and develop effective pest management strategies. An SDM mainly includes a bioclimate analysis and prediction system (BIOCLIM), a genetic algorithm for rule-set production (GARP) and a maximum entropy model (MaxEnt) [34,35]. Among them, as a representative, a MaxEnt model provided important data on predicting relationships between the distribution patterns of the FAW and climate change scenarios in recent years [36,37]. The explicit consequence of the MaxEnt model was that the risk zonings of *S. frugiperda* covered most territories of China and were divided into high (e.g., Guangxi, Fujian), medium (e.g., Yunnan, Hainan) and low (e.g., Sichuan, Anhui) suitable habitats in consideration of the known distribution area and restricting factors such as temperature and precipitation [38,39,40]. The noteworthy advantages of the MaxEnt model were a higher prediction with only a small sample size, its ease of interpretation of the prediction results, the measurement of the importance of environmental variables by the jackknife method [41,42,43]; however, some limitations were that the spatial bias of occurrences records was not well accounted for [44], and only estimates of the relative suitability were provided, taking no account of the background of the samples [45]. If the influencing factors such as land utilization rate, crop species and planting pattern could be further considered, the prediction results might be more accurate. Additionally, all our conclusions were hypothetical and need to be validated under real conditions in future studies [46].

## 3. Effects of Temperature on Phenology

Biological phenology is the scientific research of periodic biological phenomena, such as the migration of animals in relation to climate conditions, close to ecology and meteorology in biology. The data of phenological observations reflect the synthesis of climate conditions and effects on organisms, such as the insect pest response to temperature and humidity, and need to be monitored simultaneously with meteorological factors, which can be utilized to forecast the occurrence of insect pests. In this part, we discuss the relationship between representative ecological traits (life cycle, herbivory, flight capability) and temperature changes of the FAW and point out some areas that are rarely covered and worth an in-depth study, which contribute to predicting the physiological and ecological responses of the FAW in the context of future climate change, especially temperature change, and improve the effectiveness of the FAW control.

### 3.1. Life Cycle of Fall Armyworm

Temperature fluctuations affect insects’ life-cycle strategies and many data resources have assessed the direct impact of temperature on the FAW biological characteristics. The developmental stages of the FAW are divided into four stages, namely, egg, larva, pupa and adult. The periods of the egg, larval, pupal and adult stages vary with different ambient temperatures, which are about 2–3 days, 13–14 days, 7–8 days and 10 days in warm summer, respectively [47,48,49]. The periods of egg hatching is 2~4 d at 21~27 °C [50], in comparison with 1~3 d at 32~36 °C [51]. Larva, pupa and adult experience 14~17 d, 7.82~30.70 d and 14~15 d at 24~27 °C, 18~32 °C and 17~27 °C, respectively [52,53]. The overall period is 22.57~58.73 d from 17~37 °C [52,54], and it takes about 30 d to go through a generation at 28 °C [47,48,49,50,51,52,53,54,55]. The FAW will be inclined to migrate to annual breeding areas (such as Florida, southern Texas in the USA and most areas of Africa) with suitable temperature conditions [11,56]. In China, Yunnan, Guangxi, Guangdong, Hainan and other provinces (regions) are the annual suitable breeding areas of the FAW [57] (Table 1).

Temperature change is inclined to induce phenological changes in potentially invasive insects. The temperature in Florida gradually rises toward the south, which means that the number of FAW generations increases if temperatures remain at high levels, and farmers may face new challenges to cope with an increase in population numbers of the FAW [34]. The combination of accelerated developmental rates and the increased number of generations can hence lead to an expansion of the insect’s geographical range and outbreaks under higher temperatures [58,59,60]. Generally, the “temperature–size rule” refers to the fact that elevated temperatures increase metabolic rate, resulting in a higher growth rate, a shorter development time and abnormal body figures [61]. Examples occurring in *S. frugiperda* caterpillars have been proved by many scholars at high temperatures [51,52,53,54,62], but temperatures slightly to moderately increased affects body length and weight when reared on an artificial diet [25]. On the contrary, when FAW larvae are exposed to corn foliage at cooler temperatures, they reduce intakes, grow more slowly to avoid unfavorable environmental conditions, as it takes as long as 80~90 days to breed a generation, and the period is longer [63]. Over a period of time, FAW infestations occur during cooler periods and cause less damage than during warmer periods in the field [64]. *S. frugiperda* cannot, however, survive periods of extreme cold, as well as periods with mild cold and rainfall (Table 2).

The FAW adult has a strong reproductive ability; the lifespan of a female adult is generally 7–21 d. Adults can mate and lay eggs many times with 1500 eggs on average and a maximum of 2000 eggs. The FAW is nocturnal, and mating activity peaks before midnight, mainly depending on temperature [48]; the highest number of matings occurs from 20 °C to 30 °C with little mating at 10 or 15 °C [73]. The responses of virgin fall armyworm moths to changing temperature would be most significant than either single or multiple-mated moths, and they tend to mate earlier in the night [74]. However, there is no correlation with the number of matings, fertility, longevity and temperature. The average number of eggs laid by a single female is the highest at 27 °C after about 2 days of mating [51], which is consistent with the optimal temperature previously reported in terms of length and weight for this species [62].

### 3.2. Effects of Temperature on the Host Plants Selection

Omnivorous FAW populations annually infest a variety of plant species, mainly including Poaceae, Compositae and Leguminosae, feed mainly on all growth stages of corn (sweet corn and waxy corn preferably). In general, the growth parameters appear differently when FAW populations infest Poaceae over other groups of plants [17]. For instance, FAW larvae are divided into six instars and the developmental period of every instar on cotton is shorter than on corn, about 1.5~3.3 days when they feed on corn leaf tissue at 25 °C, slightly different from the 2.0~3.4 days when they feed on sweet corn kernels at 26 °C; the life cycle and survival rate of larvae are significantly lower on cotton than artificial diet or corn [53,75,76]. Host-associated genetic differentiation has subdivided the FAW into two morphologically indistinguishable strains by molecular identification (mainly by cytochrome c oxidase subunit I, *CO*I gene and triose-phosphate isomerase, *Tpi* gene), the “corn-strain” (C) primarily feeds on corn, cotton and sugarcane; the “rice-strain” (R) gives priority to feed rice and various grasses [77,78]. Although molecular identification results of the FAW populations’ samples from several provinces of China were different using molecular markers, according to the increased number of samples covered with different types of host plants and a more detailed gene fragment detection, it was concluded that the FAW population that invaded China evolved from the hybrid of an “R” female parent and a “C” male parent and the special “C” with dominant nuclear genome [27]. In addition, differences between the larval development times, wing shape and size, host plant range, sex pheromone composition, mating behavior and resistance from the “C” and “R” were documented [79,80,81,82,83].

A tight synchronization of phenology between certain herbivore species and host plants is often necessary for both to perform well [15]. The host plant can only serve as a food resource for a limited time, and in most cases, the temperature suitable for the growth of the host plant is also suitable for the development of the FAW. Temperatures that are extremely high or low for a host plant’s growth can also influence the growth of the FAW. At higher latitudes with lower temperatures, the host plant grows slowly and cannot provide sufficient food resources to support the FAW’s development, thereby reducing the digestibility and swelling rate of plants’ cellulose, while in mid/low latitudes (warmer areas), the host plant may be sufficient to provide food resources, thus accelerating the destruction of the host plant. For the FAW migratory pest, seeding fresh corns in different time and regions is conducive to the migration occurrence and a concentration of the infestation. It is relatively complex to predict future FAW’s population dynamics in relation to the host plant’s growth strategies simultaneously, and large field and laboratory studies should be conducted over a range of temperatures. The knowledge of the status of available host plants infested by the FAW under temperature change contributes to the field management and pest control.

### 3.3. Effects of Temperature on Flight Performance

This exceptional rapid invasion of the FAW can be ascribed to its long-distance dispersal behavior. Through main scientific technologies such as radar observation, trajectory analysis and meteorological analysis, it has been detected that the FAW is a typical long-distance migratory pest. Team academician Wu from the Chinese Academy of Agricultural Sciences (CAAS) predicted that this pest showed a seasonal migration from north to south in spring and summer (march to august) through the analysis of the path of a Burmese insect source into China, the falling areas and a seasonal migration rule [25,84]. Prof. Hu of Nanjing Agriculture University (NAU) reported that the FAW from northern Indochina and Myanmar could enter the annual breeding area of China and migrate to the main maize producing areas in the north through two routes, namely, east and west, based on a migration trajectory analysis and historical meteorological data [85]. Moreover, the FAW can also cross the sea from the south and east of China into Japan and Korea [86].

Nevertheless, this pest ontogeny does not have a diapause mechanism [33]. A strong flight performance, together with appropriate environmental conditions, is a key factor contributing to long-distance migration [87,88]. Inappropriate environmental temperatures force the FAW to descend to an altitude with a more suitable temperature or land, which affects migration and flight capabilities [89,90]. To survive in the low temperatures of winter in the Americas, FAW adults migrate south for warmer climates and then reinvade throughout the USA and into Canada the following summer [91] or migrate from the western parts of the African continent to southern Sahara in one single night or over a few consecutive nights [92].

There is only restricted information on quantifying the relationship between temperature and flight performance in this species. The first comprehensive radar-based monitoring of *S. frugiperda*’s flight performance using a flight mill apparatus revealed that all flight parameters initially increased and then gradually dropped at 10~30 °C, inherently possessed a strong flight ability at 20~25 °C and a low temperature, especially 10 °C, could obviously reduce flight speed and wingbeat frequency [93]. Based on the logit regression, the threshold temperatures of flight of *S. frugiperda* were 14.9 °C and 13.1 °C at 10~20 °C and 20~10 °C, respectively [94]. However, *S. frugiperda* adults showed the highest flight performance at 32 °C, which is unfavorable for development but favorable for escaping from a detrimental environment [66]. Additionally, by adjusting flight time and speed, the trial insect could achieve the same flight distance, which could correspond with an energy metabolism that is slow at low temperature to meet the energy supply for longer flight and thus achieve longer flight distance.

## 4. Effect of Temperature on the Biological Agents as Representation

Many reviews have summarized in detail and comprehensively the control techniques of the FAW, including agricultural, biological, chemical control and monitoring techniques. The Ministry of Agriculture and Rural Affairs, China (MARA), NATESC and other relevant departments have proposed that the strategy of “regional management, joint prevention and control, comprehensive management” be implemented, addressing the ecological control and agricultural control as the basis, biological control and physical and chemical control as the focus and chemical control as the bottom line [95]. Biological control agents (natural enemy resources such as parasitoids, predators and entomopathogens) show environmentally friendly and sustainable advantages, present positive application prospects in the management of *S. frugiperda*, while chemical and other controls, such as the application of synthetic insecticides, are accompanied by environmental contamination and resistance development. Therefore, this paper mainly reviews the relationship between biological agents and temperature. In total, 250 natural enemy insect resources (206 parasitoids and 44 predators) of *S. frugiperda* giving priority to *Hymenoptera* and *Diptera* [96], 47 entomopathogens resources centered on *Hemiptera* and *Coleoptera* were summarized [97]. Here, we just listed several parasitoids, predators and entomopathogens of *S. frugiperda* and the effects of temperature on their growth and control effect, providing critical information for the development and implementation of enhanced biological control programs.

Many parasitoids possess strong parasitic ability for *S. frugiperda* [72,98,99], such as *Telenomus remus* Nixon and *Trichogramma pretiosum* Riley [100]. There is an inverse correlation between developmental time and temperature at 15~34 °C [101,102]. To ensure a high parasitic efficiency, the time of parasitoids release should be early in the morning to allow adults to find shelter from high temperatures, avoiding the late evening because the parasitoid is inactive at that time [103,104]. Of note, the sex ratio and number of parasitoids per egg are not influenced by temperature (this may be related to the number of host available and age of the parasitoids females) [105,106], thus ensuring the quality of laboratory insect mass production to meet the needs of a large-scale release on field. In the case of predators, predation efficiency varies with temperature and age; for instance, in *Orius sauteri* Poppius, the highest instantaneous attack rate and predation quantity occurred predominantly at low instar larvae especially the first instar at 25 °C and the predatory efficiency of *Eocanthecona furcellata* Wolff for low instar larvae of the FAW gradually increased at 17~32 °C and was optimal at 32 °C [107]. In conclusion, the comprehensive considerations of field environmental factors and the larval instars of *S. frugiperda* are vital for later successful field releases, including times, rates and frequencies [108].

Entomopathogen resources of *S. frugiperda* are relatively abundant including bacterium, virus, fungi, microsporidia and nematode. In the Americas, local farmers sometimes collect dead and dying larvae (the body is filled with virus particles or other pathogens that are in the stage of infection). After grinding and filtering, filtrate containing virus or fungus and water are mixed, then they are sprayed in the field, especially where the plants are being infested [109]. Biological products made from entomopathogen e.g., *Bacillus thuringiensis* (Bt), Beauveria bassiana, *Metarhizium anisopliae* and nucleopolyhedrovirus (NPV) have been approved for the control of *S. frugiperda* in China [47,110,111,112,113,114]. Entomopathogens can modulate the physiology, ecology and behavior of the host to maximize their own reproduction and adaptability to the environment [115,116,117,118,119]. Appropriate temperature is the prerequisite for successful control and has a direct influence on the parasitism and infectivity of nematodes, for Noctuidonema guyanense Remillet & Silvain, 32 °C is the most suitable for parasitoid population growth [120,121] and is also within the favorable range for the FAW’s growth. The effect of other temperatures on the overall abundance of FAW populations remains unknown due to a lack of equivalent data on natural enemies. Not only the own reproductive capacity of insect pest populations, but also the abundance of predators and parasitoids decide the population density of insect pests [122]. In the case of a more rapid expansion of the FAW over natural enemies, it will undoubtedly cause more outbreaks.

## 5. Temperature Tolerance and Regulating Genes

The capability of temperature tolerance of an invasive species is interpreted to be a crucial challenge in its successful establishment and dissemination under extreme climate change [123]. In general, within a certain temperature range, especially when it comes to inappropriate temperature regions for development, some ectotherms could maintain cell homeostasis and improve the body’s defense ability by improving gene and protein levels; however, a weak development or extinction damage will be shown when the ambient temperature exceeds the insect’s natural tolerated environmental limit and even a brief exposure to extreme temperatures could affect population dynamics, organism phenology and community structure [30,124,125].

A successful adaptation to environmental temperatures can promote opportunities to complete its life cycle and is also an important reason for the rapid outbreak of the FAW globally. However, the literature about the heat hardiness of the FAW remains poorly elucidated; although modern molecular techniques may soon allow rapid progress, studies have only reported that the population growth is inhibited at extreme high temperature [67]. Similarly, for other pest species, the survival and reproduction of *B. tabaci*, *S. avenae*, *M. persicae* and *E. postvittana* were inhibited at extremely high temperatures [68,69,70,71,72]. All of the developmental life stages of the FAW withstand low temperatures to varying degrees: egg < adult < larva < pupa [126]. However, the adult stage was also reported to be the most sensitive stage to cold temperatures, with only 25% surviving at ~5 °C and death at colder temperatures, while the most tolerant stage, the egg stage, had a 30% survival at ~10 °C, which violates our conclusion from the discussion above; we suspect that different measurement and analysis methods led to the two different results [65] (Table 2). Insects can be divided into freezing-susceptible and freezing-tolerant according to supercooling points (SCP) [84,126,127,128]. However, most studies revealed that populations from different regions, insects infesting different host plants and SCP determination methods could affect the results of SCP, resulting in the SCP not being an adequate predictor of cold hardiness [129,130].

To resist temperature stresses, insects will make corresponding physiological adjustments, including gene expression, aimed to improve the viability ability under environmental stress. Studies on heat shock protein genes (e.g., *Hsc70*, *Hsp90*, *sHsp19.07*, *sHsp20.7* and *sHsp19.74*) and *trehalose-6-phosphatesynthase (TPS)* genes (*CYP4G15* and *CYP4L4*) have shown that they participate in the regulation of the tolerance of FAW to temperature fluctuations [131,132,133,134,135]. The heat shock response was first discovered by Italian geneticist Fernando Ritossa, when he observed a new buffing pattern on the salivary gland chromosomes of Drosophila as a response to increasing temperature and exposure to certain chemicals [136]. The tolerance proteins are products of different environmental stress (such as low/high temperature) and enhance the organism’s resistance for a better survival [137,138,139,140]. In addition, the expression levels of Hsps genes differed among the developmental stages (larva, adult) and tissues (ovary, abdomen, head, compound eyes, antennae) of male and female adults [139,141], suggesting that these are related to the external environmental stimuli, requiring a large number of proteins to assist cell transport of chemical signals. Research on temperature tolerance is instrumental for decoding the molecular mechanism behind its wide adaptation ability across different regions and helping us develop better control management.

## 6. Management Strategies in the Background of Global Warming

Currently, synthetic insecticides are most frequently used to control FAW, however, the overuse and continuous repeated spraying of synthetic chemicals is not only ineffective, uneconomical and unsustainable, but also poses health and environmental hazards to farmers, consumers and the ecosystem [142]. Moreover, host plant resistance, the abundance of natural enemies and the effectiveness of synthetic chemicals used for pest management will be reduced with changes in global temperature [143]. Integrated pest management (IPM), known as a robust construct to arise in agricultural sciences, encourages the users or producers to take full advantage of available optimal pest control options within ecological systems in consideration of economic, environmental and social benefits. Authorities have recommended climate-smart pest management (CSPM) strategies, including the combination of local climate forecasting and actual changing observations together with a pest risk assessment into pest management planning strategies [144]. Together with the reduction in pest-induced crop losses that CSPM brings a double-win effect of adapting to evolving pest threats and ultimately raising food security. Given this, there are components of IPM that could enable a sustainable control of the FAW after climate change, as discussed below.

### 6.1. Monitoring and Early Warning

At the country and community levels, we propose that the public and private sectors establish an integrated system of pest and climate conditions forecasting, surveillance, detection and control based on CSPM, making farming systems more resilient to climate, e.g., temperature change, reducing the negative impacts on the broader ecosystem. Methods mainly involve field reconnaissance, releasing pheromone traps, mass effective data collection and creating databases, helping uncover the development–temperature relationship of insect pests’ models and creating data visualization tools contributing to predict possible future outbreaks [145]. The FAO has recommended that early warnings consist of a centralized cloud-based platform comprising a global database linked to a geographic information system (GIS). More awareness should also be raised by governments to enact relevant policies and regulations to speed up the evaluation, registration and quality management of the fall armyworm management options. We put forward some control suggestions and recommended measures, which should be implemented by relevant departments including federal and local authorities, research institutes, major agricultural stakeholders and farmers so the battle between man and the FAW is successful.

### 6.2. Suitable Cropping System

A cropping system describes how crop patterns can maximize the rational use of land resources, reduce disease and insect pests and improve output. The aim is to promote a crop’s healthy growth and improve the crop’s resistance through strengthening field management, rational fertilization and watering. According to temperature conditions, the sowing date of crops can also be adjusted to stagger the sensitive growth period and the occurrence period of insect pest and diseases to minimize the suitable habitat for insect pests. CSPM puts forward that the strategy of minimum tillage and planting natural barriers not only increases organic carbon sequestration in soil, but also increases the resilience to certain pests. Identifying and introducing plants that house natural enemies and facilitate their reproduction is also a simple and cost-effective alternative. The rotation of crops repelling pests or attracting natural enemies reduces food source and oviposition on the host plant, which may significantly reduce recurrent pest infestations each growing season. For instance, the “push-pull” companion cropping strategy means interplanting crops to drive the FAW (“pull”) and plant weed traps nearby to attract the FAW to the surrounding weeds (“push”); it has been proved effective in reducing the number of larvae per corn plant (82.7%), decreasing damage degree (86.7%) and increasing yield (2.7 times) in Kenya, Uganda and Tanzania [146]. Governments should also encourage the growers to modify crop planting plans for plants that are vulnerable to be attacked by the FAW at certain temperature and even provide financial support to the modification and take strict measures of prevention and control to prevent the invasion of this pest to a continent of low suitability.

### 6.3. Use of Biological Control

As an ecologically friendly means, a biological agent would be an ideal alternate for effectively combating and sustainably managing the *S. frugiperda*. However, different thermal preferences between crop pests and their natural enemies may lead to a loss of synchronization between the two biologies and increased risks of pest outbreaks [147,148]. Additionally, for other biological control agents, due to the development of high resistance levels for Bt transgenic events in recent years [149,150], vegetative insecticidal proteins (Vips) are considerable alternatives of resistance management strategy and have a specific insecticidal activity to lepidopteran pests resistant to Cry insecticidal proteins and promising no cross-resistance with Cry proteins [151,152]. When the biological products are used in practice, attention should be paid to the host plant’s growth status and the occurrence of insect pests under temperature conditions in the field and the release of biological products reasonably should be aimed to maximize their effectiveness.

The global climate conditions are complex; we should timely adjust measures according to the climatic environment in different regions and crop planting patterns, establish an integrated prevention and control technology system on the basis of the ecological regulation, with biological control as the core and the emergency prevention and control of chemical pesticides as a complementary measure, in order to minimize hazard losses and ensure the safety of national food production.

## 7. Conclusions

Summarizing, future climatic change will affect the FAW’s occurrence in different ways and degrees; higher temperatures, all other things remaining fixed and equal, allow elevated development rates, probably additional generations within the same time, then an earlier migration into wider geographical ranges, infesting more host plants. Warmer winters may lead to a greater survival, advancing the first flight threshold of the year, but may result in a lower adult weight and fecundity. On the contrary, low temperature prevents essential movement to new feeding sites, leading to death due to starvation. The tolerance responses of the FAW to temperature stress could mitigate the adverse effects of climatic change. In this review, we discussed that temperature directly influenced the geographic distribution, life cycle and flight performance, or indirectly influenced the development of host plants and natural enemies of the poikilothermal FAW, thus may affect the damage to host plants. We also summarized the partial temperature tolerance mechanism, including genes and proteins expression, and put forward some control strategies in the background of future climate change (focusing on temperature).

What is noteworthy is that detailed laboratory experiments are usually designed in a way that allows the biological behavioral response to the variation of one element only and keep the others as constant as possible, which accelerates the experimental process of simulating actual conditions or operating under field temperature stress [153]. However, the thermal performance of insect pests at fluctuating temperatures may differ from those at constant temperatures set up in lab conditions [124]. The outbreak of FAWs is not out of mere coincidence, but a product of being driven by multiple factors [154]. Climate change increases the likelihood of other extreme events (e.g., droughts) that needs to be emphasized. The interactions between abiotic and biotic and direct and indirect components may prevent the determination of community-level consequences.

The influences of climate change on the FAW’s manifestation are complex; there are still several unknowns that were not illustrated by this review: (i) Although FAW populations are subdivided into two morphologically indistinguishable strains (“R” and “C”) by the *CO*I and *Tpi* genes based on a host-associated genetic differentiation, the difference between the two strains’ response to temperature change remains unknown; thus, a further monitoring of the “R” strain occurrence and trends in the evolution of the two strains in China and finding strategies to cope with ambient temperature adaptation, is necessary; (ii) Although direct effects and indirect ones are difficult to disentangle, currently, little is known as to how these may change in the near future and they warrant a discussion. A comprehensive analysis should also be carried out based on the considerably more detailed investigation of actual field and climatic conditions, aiming to provide more comprehensive and reliable data support for the division and prediction of the damaging region of *S. frugiperda* in the near future; (iii) When considering that temperature acclimation increases the tolerance of S. *frugiperda*, it is necessary to further investigate whether the genes that are induced by cold/heat stress are significantly upregulated during recovery from cold/heat shock and their variations at the protein and metabolism levels. Moreover, the underlying mechanism explaining how other proteins counteract stressful temperatures in this invasive pest remains murky and needs further refinements.

Research on the effects of climate change on the FAW, especially temperature, still has many challenges, including formulating the unknowns mentioned above; for future research, we suggest considering the following points:Paying close attention to global climatic change, developing international cooperation and improving capacities of forecasting and surveillance, making efforts for controlling the FAW globally;Multiple factors including limits of natural enemies and host plants need to be considered when assessing the effects of temperature on the FAW’s dynamics, putting together a wider context. Additionally, abiotic factors such as precipitation might alter relative humidity and is likely to affect important physiological functions, e.g., reproduction, which indirectly affects the direct effect of temperature on the FAW;The occurrence of the FAW around the world will be normal and long-term prevention and control strategies should be adopted in annual breeding areas. Information about long-term FAW population-level variation to global climatic change is scarce, so this context should be further strengthened.

## Figures and Tables

**Table 1 insects-13-00981-t001:** Developmental periods of FAW at various temperature.

Stage	Temperature (°C)	Period (d)	References
Egg hatch	21~27	2~4	[50]
Egg hatch	32~36	1~3	[51]
Larva	24~27	14~17	[53]
Pupa	18~32	7.82~30.70	[53]
Adult	17~27	14~15	[52]
Overall development	20~35	23.0~48.3	[54]
Overall development	17~37	22.57~58.73	[52]
Overall development	28	30	[55]

**Table 2 insects-13-00981-t002:** The detrimental effect of temperatures on FAW and other species.

Species	Temperature (°C)	State of FAW	References
*S. frugiperda*	32~36	Developmental period was shortened with increasing temperature	[51]
*S. frugiperda*	17~32	Developmental period was shortened with increasing temperature	[52]
*S. frugiperda*	18~32	Developmental period was shortened with increasing temperature	[53]
*S. frugiperda*	20~35	Developmental period was shortened with increasing temperature	[54]
*S. frugiperda*	23~31	Developmental period was shortened with increasing temperature	[62]
*S. frugiperda*	Below 5	Lethal effect at extreme low temperature	[65]
*S. frugiperda*	Below 10	Lethal effect at extreme low temperature	[30]
*S. frugiperda*	Below 20 Exceed 36	Flight capability was reduced	[66]
*S. frugiperda*	Above 38	Growth was inhibited	[67]
*Sitobion avenae* (Fabricius)	31	Reproduction was decreased	[68]
*Mysus persicae* (Sulzer)	37	Developmental rate was decreased	[69]
*Epiphyas postvittana* (Walker)	Above 40.4	Lethal effect at extremely high temperature	[70]
*Bemisia tabaci* (Gennadias)	44	Growth was inhibited	[71]
*Oryzaephilus surinamensis* (Linne)	36~48	Developmental rate and fecundity were decreased	[72]

## Data Availability

All data presented in this study are available in the article.

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
