# Peer review of "Impact of Temperature Change on the Fall Armyworm, Spodoptera frugiperda under Global Climate Change"

_insects, 2022, doi:10.3390/insects13110981_

Round 1
Reviewer 1 Report
The intention of the authors, as stated in the abstract, is to provide a review of the potential impacts of climate change on the life-history, population dynamics and distribution of Spodoptera frugiperda.
The theme might be of high relevance as climate change are expected to affect physiological responses, dynamics and distribution of relevant pests such as the fall armyworm.
However, the authors fails in providing a clear and logic structure of the review manuscript. As it stands, the manuscript provide a set of jeopardised information about some aspects linked to climate change and the pest. Elements provided are not always supported by relevant literature and they are hastily presented and discussed.
Literature sources about the role of temperature on FAW are vast and the review manuscript fails in providing a clear, informative and well-discussed summary of the main scientific results available respect to the topics that are presented.
The objectives (and the limits) of the review must be carefully defined, as well as the structure of the manuscript.
The english language used makes the whole manuscript hardly understandable. The jargon used is not scientifically-sound and, in some cases, major issues in the explanation of the basic biological concepts are present (sometimes, biological concepts are wrong, at least in their explanation).
I tried to provide comments (see attachment) about the major issues I have found.
Based on this review, I consider the manuscript not suitable for publication,

Author Response
Thank you for all kind comments and suggestions, we revised structure of the review manuscript, especially some hardly understandable paragraphs and sentences, we defined carefully the objectives, updated the relevant literature and added discussion section to make the the review more understandable. For each comment, please see the attachment.

Reviewer 2 Report
Manuscript Title: C Impact of Temperature Change on the Fall Armyworm, Spodoptera frugiperda under Global Climate Change
Manuscript insects-1963937 contributes give the valuable information to the researchers and readers. The subject of the manuscript is consistent with the scope of the Journal. Thus, I suggested that the manuscript need to be major revised before it is accepted by this journal.
1. The manuscript (MS) let down by poor language. This drawback cause the perceived inaccurate and imperfect scientific style throughout all the MS sections preventing the readers to observe the MS novelty and challenging content. I could recommend to the editor that the authors have their paper language edited before resubmission, if applicable.
2. Title: Fall Armyworm is italicized, the text is not?
3. Line 56 : change “Much” to “much”.
4. Line 59 : The extra space should be removed
5. Line 72, 75, etc.: “Zea mays” should be italicized.
6. Line 56 : change “method” to “methods”.
7. The introduction and conclusion are extremely weak.
Author Response
Thank you for all kind comments and suggestions, we edited the review language before resubmission, revised structure of the review manuscript, especially some hardly understandable paragraphs and sentences, we defined carefully the objectives, updated the relevant literature and added discussion section to make the the review more understandable. All comments were responded carefully, please see the attachment.

Round 2
Reviewer 2 Report
I have no comments.